# A Mobile Solution for Enhancing Tourist Safety in Warm and Humid Destinations

**Sairoong Dinkoksung [1], Rapeepan Pitakaso [2], Chawis Boonmee [3], Thanatkit Srichok [2,*], Surajet Khonjun [2], Ganokgarn Jirasirilerd [4], Ponglert Songkaphet [5] and Natthapong Nanthasamroeng [6]**

[1] Department of Hotel Management, Faculty of Management Science, Ubon Ratchathani University, Ubon Ratchathani 34190, Thailand; sairoong.d@ubu.ac.th

[2] AIO SMART Laboratory, Department of Industrial Engineer, Faculty of Engineering, Ubon Ratchathani University, Ubon Ratchathani 34190, Thailand; rapeepan.p@ubu.ac.th (R.P.); surajet.k@ubu.ac.th (S.K.)

[3] Department of Industrial Engineering, Faculty of Engineering, Chiang Mai University, Chiang Mai 50200, Thailand; chawis.boonmee@cmu.ac.th

[4] Department of Industrial Management Technology, Faculty of Liberal Arts and Sciences, Sisaket Rajabhat University, Sisaket 33000, Thailand; ganokgarn.j@sskru.ac.th

[5] Department of Computer Science, Faculty of Computer Science, Ubon Ratchathani Rajabhat University, Ubon Ratchathani 34000, Thailand; ponglert.s@ubru.ac.th

[6] Department of Engineering Technology, Faculty of Industrial Technology, Ubon Ratchathani Rajabhat University, Ubon Ratchathani 34000, Thailand; natthapong.n@ubru.ac.th

* Correspondence: thanatkij.s@ubu.ac.th

**Abstract:** This research introduces a mobile application specifically designed to enhance tourist safety in warm and humid destinations. The proposed solution integrates advanced functionalities, including a comprehensive warning system, health recommendations, and a life rescue system. The study showcases the exceptional effectiveness of the implemented system, consistently providing tourists with precise and timely weather and safety information. Notably, the system achieves an impressive average accuracy rate of 100%, coupled with an astonishingly rapid response time of just 0.001 s. Furthermore, the research explores the correlation between the System Usability Scale (SUS) score and tourist engagement and loyalty. The findings reveal a positive relationship between the SUS score and the level of tourist engagement and loyalty. The proposed mobile solution holds significant potential for enhancing the safety and comfort of tourists in hot and humid climates, thereby making a noteworthy contribution to the advancement of the tourism business in smart cities.

**Keywords:** tourism safety; tourist loyalty; SUS; mobile application; intelligence tourism

## 1. Introduction

Preserving the safety of tourists in hot climates is of utmost importance, considering the potential dangers associated with heat-related illnesses, notably heat exhaustion and heatstroke [1]. While recent studies have examined strategies to promote tourist safety [2,3], additional efforts are required to minimize the impact of heat-related illnesses. Tourists should prioritize preventive measures, including maintaining hydration, minimizing sun exposure, and taking breaks in shaded areas [4]. Thorough research on travel destinations is also vital to identify potential heat-related hazards [5,6]. By prioritizing safety, the tourism industry can effectively manage heat-related risks and ensure safer travel experiences [7].

Tian et al. [8] underscored the importance of thermal comfort in urban tourism environments, exploring the impact of diverse factors, including physical, individual, societal, and psychological aspects. Their research highlighted the importance of determining suitable environmental conditions for tourists to promote safer visits. These factors exhibit seasonal variations, with physical factors playing a dominant role, suggesting the potential to utilize current weather conditions to mitigate heat-related illnesses. Therefore, providing tourists with up-to-date weather conditions and information about heat-related

illness risks at tourist attractions would decrease their chances of experiencing heat-related illnesses [9–13].

To overcome the limitations of existing tourist information, our study aims to develop a feature-rich mobile application that acts as a comprehensive platform for providing vital information on heat-related illnesses to tourists. The application will include a warning system that notifies tourists about relevant details, such as humidity, weather forecasts, attraction conditions, and temperature. Additionally, it will monitor tourists' heat-related illness indicators and incorporate a rescue system, connecting them with nearby medical centers to ensure their safety while visiting hot and humid attractions.

The research objective of this study was derived through careful consideration of the current challenges faced by tourists in warm and humid destinations, as well as the existing solutions available in the previous market. After conducting a thorough literature review and analyzing the strengths and weaknesses of the available solutions, we identified a gap in the market for a comprehensive mobile application that integrates weather and safety information, health recommendations, and a life rescue system. Therefore, the research objective of this study is to develop a mobile solution that addresses this gap and enhances tourist safety in warm and humid destinations.

Mobile applications have evolved into indispensable instruments for furnishing tourists with health and safety information [14–16]. Noteworthy research studies conducted by Ngeoywijit et al. [17], Aiello et al. [18], and Pranoto et al. [19] have recently investigated the potential of mobile applications in augmenting tourist safety within urban locales. These studies emphasize the importance of establishing trust, adopting user-centered development approaches, and delivering reliable information to improve user adoption and engagement [20–22]. Mobile applications play a critical role in enhancing tourist safety within the tourism industry.

Our objective is to create a unique mobile application that surpasses existing literature-based applications by providing a warning system with crucial weather information (humidity, UV status, and temperature) and details about destination services that mitigate risks like dehydration, heart attack, and heatstroke. The application will also assess the risk of heatstroke based on smartwatch data, generate timely reports on abnormal conditions, and establish seamless communication with nearby rescue organizations to ensure immediate assistance for tourists in need. Furthermore, hospitals will have real-time access to patients' locations within their service area. Our ultimate goal is to enhance tourist safety and security, particularly in larger areas with multiple destinations.

The tourism industry is facing a growing need for effective weather and safety information systems that can address the unique challenges of warm and humid destinations. Existing literature-based applications have limitations in providing comprehensive and reliable information to tourists, which can lead to safety concerns and negative experiences. To address these limitations, our objective is to create a unique mobile application that integrates weather and safety information, health recommendations, and a life rescue system to enhance tourist safety in warm and humid destinations.

The proposed mobile application will provide a warning system with crucial weather information (humidity, UV status, and temperature) and details about destination services that mitigate risks like dehydration, heart attack, and heatstroke. It will also assess the risk of heatstroke based on smartwatch data, generate timely reports on abnormal conditions, and establish seamless communication with nearby rescue organizations to ensure immediate assistance for tourists in need. Furthermore, hospitals will have real-time access to patients' locations within their service area. By addressing the limitations of existing tourist information and providing a comprehensive solution that meets the needs and concerns of tourists in warm and humid destinations, our research aims to make a significant contribution to the field of tourism and tourist safety.

Based on the comprehensive literature review conducted, the ensuing topics constitute the primary focal points of our research contributions.

1.  The development of a comprehensive mobile application that integrates weather and safety information, health recommendations, and a life rescue system to enhance tourist safety in warm and humid destinations.
2.  The incorporation of three distinct input datasets from various sources, including internet-based data, data obtained from a custom-designed application linked to tourists' smartwatches, and data directly collected from tourist attractions, to facilitate communication between tourists and emergency service providers in the relevant areas.
3.  The proposal of a research methodology consisting of two main components: data collection and the design of the tourist safety system, which can be used as a framework for future research in this area.
4.  The potential to improve the management and policy of tourism and the development and implementation of highly effective weather and safety information systems for tourists through the proposed solution.

The article follows a structured organization outlined below; Section 2 presents a comprehensive literature review. Section 3 details the research methodology, while Section 4 encompasses the computational results and framework. Section 5 offers an insightful discussion, drawing comparisons between the current research and existing methodologies. Finally, Section 6 concludes the study and suggests potential directions for future research.

## 2. Literature Review

### 2.1. Cross-Cultural Perspectives on Health and Safety Practices in the Hospitality Industry

Numerous studies have extensively explored risk management practices implemented in renowned tourist destinations, underscoring the paramount importance of health and safety measures [23]. Additionally, research has diligently examined tourists' perceptions of health risks in popular destinations and the intricate factors influencing their behaviors in relation to these risks [24]. Specific risk management strategies targeting the safety of adventure tourism, particularly in activities such as zip-lining, have been comprehensively investigated [25]. Moreover, cross-cultural perspectives on health and safety practices within the hospitality industry have been thoroughly examined, delving into the challenges and exemplary approaches to managing health and safety in culturally diverse contexts [26,27]. Detailed analyses of visitor behavior and safety protocols at tourist attractions have yielded valuable insights for effective risk management [28]. Furthermore, research has examined the information-seeking patterns of travelers during pandemics, including the H1N1 pandemic, and their implications for risk management within the tourism industry [14,29]. Comprehensive systematic literature reviews have offered valuable insights into managing risks and crises in the tourism and hospitality industry, encompassing both best practices and the challenges that may arise [30]. "The concept of safety in tourism has been extensively applied in diverse facets, including the regulation of cruise ship passengers [31] and the surveillance of seawater conditions [32]".

### 2.2. Significance of Mobile Applications in Risk Avoidance and Tourism Safety

The available literature underscores the significance of effective risk management strategies for safeguarding the health and safety of tourists. However, a notable research gap exists in the development of holistic mobile applications tailored to address the specific requirements of warning tourists about the risks of a heat-related illness at their intended attractions, monitoring their well-being while on-site, and providing prompt rescue assistance during urgent situations.

While scholars have explored the utilization of mobile and web-based applications to enhance tourist safety across various destinations, their focus has predominantly centered on crisis communication, preparedness for natural disasters, and urban safety [20,33–38]. Some attention has also been given to the integration of smartwatches and wearable devices in managing tourist risks and safety, facilitating real-time monitoring, personalized recommendations, risk assessment, and emergency support [32,39–43]. In the realm of

tourism app development, researchers [44] delved into crafting an interactive design that uniquely catered to enhancing safety measures at various tourist destinations. Their innovative approach not only focused on preventive strategies but also integrated crucial first-aid assistance to promptly address any potential criminal incidents.

Furthermore, in the pursuit of ensuring traveler safety on the road, Akter et al. [45] introduced a revolutionary mobile application named "Journey Safety". This cutting-edge app aims to offer travelers comprehensive safety provisions and timely aid throughout their journeys, guaranteeing a secure and worry-free travel experience like never before.

These investigations underscore the immense potential of technology-driven solutions in bolstering tourist safety and enhancing their overall experience. However, to address the research gap effectively, further exploration and advancement are warranted within the tourism industry to develop all-encompassing mobile applications that cater specifically to the aforementioned needs of warning, monitoring, and rescuing tourists regarding heat-related illness risks. This endeavor would considerably contribute to the advancement of tourist safety measures, ensuring a more gratifying and secure experience, especially in unfamiliar environments.

### 2.3. Age and Gender Effects on Tourist Satisfaction, Engagement, and Loyalty in Mobile App Usage

The integration of mobile applications has revolutionized the tourism industry, transforming how tourists plan, navigate, and engage with destinations [42]. Nevertheless, our understanding of how age and gender influence tourist satisfaction, engagement, and loyalty in relation to mobile app usage remains limited [19,46]. Younger individuals exhibit higher levels of mobile app usage, seeking interactive and immersive experiences, while older tourists face barriers like technological unfamiliarity and privacy concerns [47].

Gender differences exist, with females inclined towards social and communication-oriented apps and males leaning towards gaming or utility-based applications [48]. Tourist satisfaction, engagement, and loyalty play vital roles in shaping overall travel experiences [49]. Yet, limited research has specifically examined the relationship between mobile app usage and these outcomes, particularly considering age and gender as moderating factors. Further exploration is warranted to comprehensively understand how different age groups and genders respond to mobile app usage and its impact on their satisfaction, engagement, and loyalty toward tourist attractions.

While direct studies on the relationship between SUS scores and tourist engagement and loyalty are limited, findings from related fields suggest that higher SUS scores enhance user satisfaction, trust, and loyalty [50]. Age and gender differences are observed, with younger users exhibiting higher levels of engagement and satisfaction and gender variations influencing user outcomes. However, further research is needed to investigate the combined effects of SUS scores, age, and gender on tourist engagement and loyalty within the tourism industry's mobile app usage context.

### 3. Research Method

The research was initiated through the administration of surveys to collect relevant information on tourist safety, focusing on heat stroke, heart attack, and dehydration. Two sources were utilized to acquire the essential data: (1) internet-based literature and (2) interviews conducted with healthcare professionals. Subsequently, a system design was devised to effectively disseminate this information to tourists, aiming to enhance their awareness and knowledge regarding the associated risks of heat stroke, heart attack, and dehydration—moreover, the system endeavors to furnish travelers with the requisite information to safeguard themselves during their journeys.

### 3.1. Methodology for Identifying Factors for Tourists to Prepare against Dehydration, Heat Stroke, and Heart Attack before Entering Attractions

The data collection phase employed a mixed-methods approach to gather information from both primary and secondary sources. The initial step involved conducting an extensive

review of pertinent literature concerning heat-related illnesses and associated risk factors. This review encompassed internet sources, academic databases, and medical journals. Furthermore, semi-structured interviews were conducted with medical professionals to gain valuable insights into the risk factors and preventive measures associated with heat-related illnesses. Secondary sources included references to Anderson and Bell [51], Epstein and Yanovich [52], Faurie et al. [53], Solan [54], and the National Institute for Occupational Safety and Health [55]. The interviews involved five medical staff members from Ubon Ratchathani University to validate the findings from these references [51–57].

### *3.2. Tourism Safety System Design Method*

In the system design phase, researchers utilized the gathered data to develop a user-friendly and interactive app aimed at providing tourists with essential knowledge and tools to safeguard themselves against heat-related illnesses. A survey was administered to a cohort of visitors exploring various destinations to assess its efficacy. The design framework of the Tourist Safety Support System (TSSS: T3S) is illustrated in Figure 1.

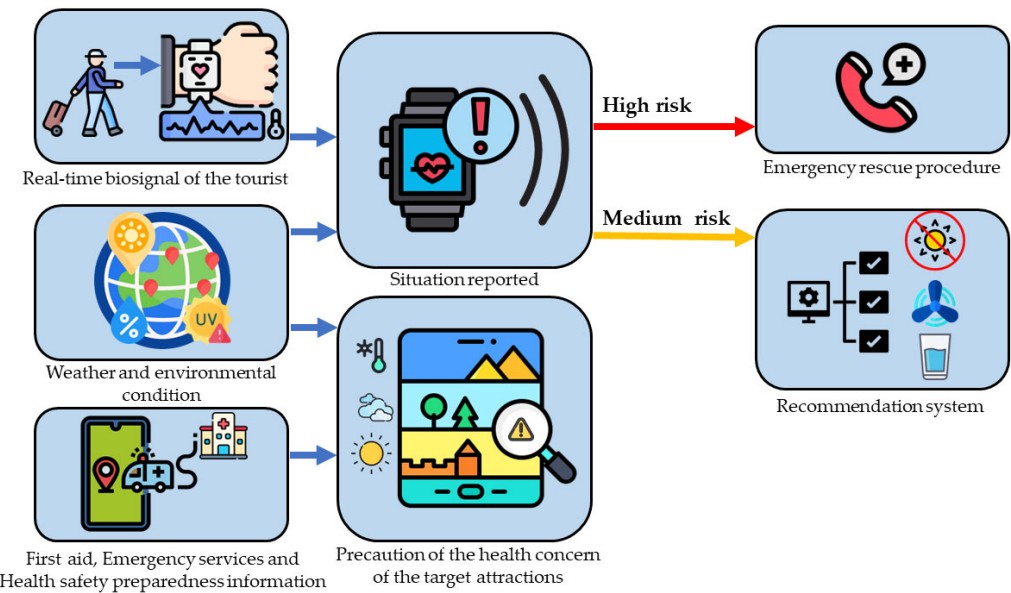

**Figure 1.** Framework of the tourism safety system. The system incorporates three distinct input datasets from various sources. These sources encompass internet-based data, such as weather and environmental conditions, data obtained from a custom-designed application linked to tourists' smartwatches, and data directly collected from tourist attractions. These datasets serve as the foundation for facilitating communication between tourists and emergency service providers in the relevant areas. The research methodology consists of two main components: data collection and the design of the tourist safety system. The data collection phase is further divided into three segments, involving the collection of data from tourist attractions, internet sources, and tourists themselves through dedicated applications and smartwatches.

#### 3.2.1. Data Collected from Tourist Attractions

The site visits were conducted for systematic observation of tourist behavior and safety practices. Trained observers were strategically stationed at different points within the attractions to identify safety risks and assess the effectiveness of existing safety measures. Additionally, interviews with attraction staff provided valuable insights into safety concerns and encountered incidents. A checklist, adapted from reputable sources such as Castleberry [58], Stanley (n.d.) [59], Virginia Tech [60], and the World Health Organization [61], was utilized to assess the presence of safety-supporting tools, activities, and locations in the observed attractions (Figure 2). Through a combination of observational methods and interviews, our study offers valuable insights into safety practices at tourist

attractions, highlighting areas for improvement in the implementation of tourist safety protection systems.

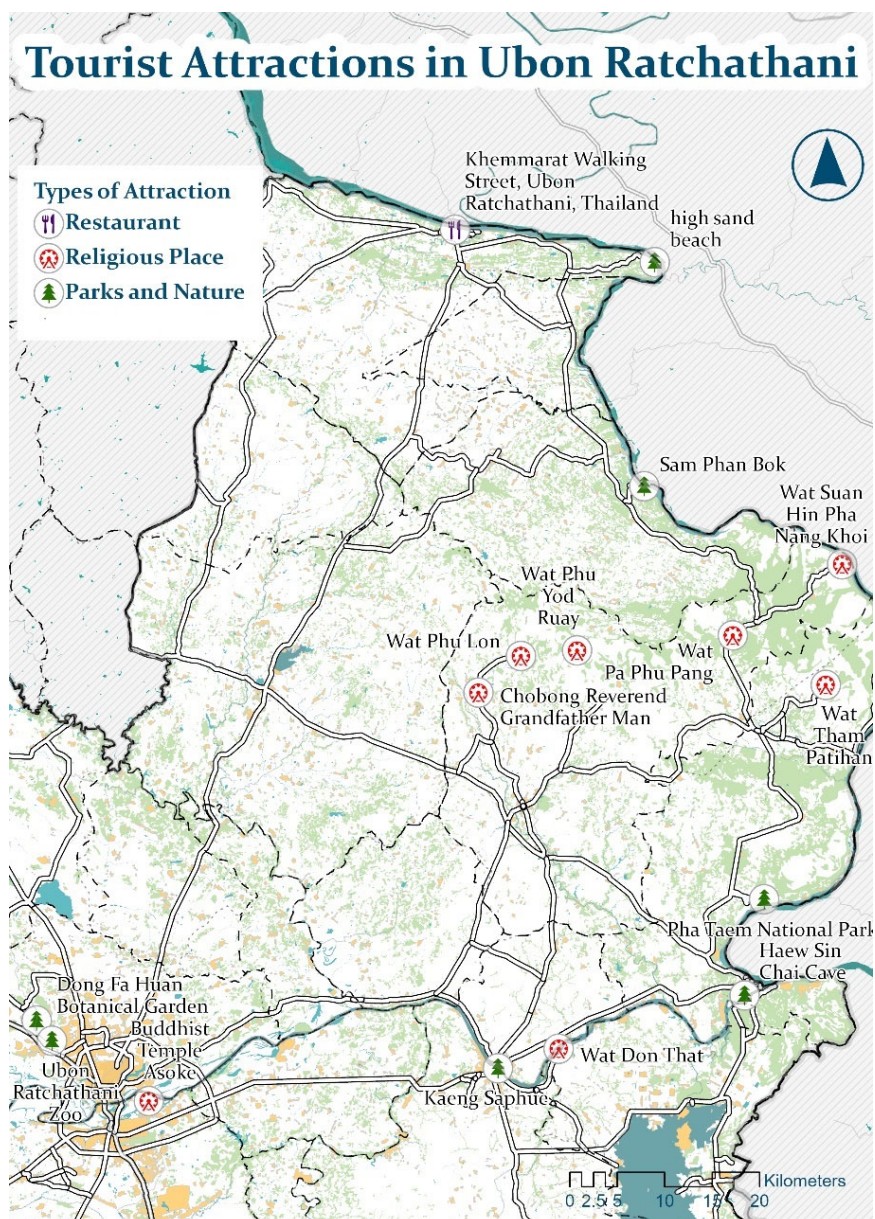

**Figure 2.** Example of the tourist attractions used in the research.

Table 1 was the checklist utilized in this research summarizes and adapts information on tools, locations, and activities from reputable sources like Castleberry [58], Stanley (n.d.) [59], Virginia Tech [60], and the World Health Organization [61]. These references provide practical recommendations for maintaining proper hydration during outdoor activities and extreme weather conditions. For example, the Stanley 1913 website offers a comprehensive guide on staying hydrated during outdoor adventures, and the World Health Organization website provides advice on staying cool and protecting oneself from direct sunlight during heatwaves. SheKnows offers tips on preventing travel dehydration, and Newswise provides outdoor safety tips emphasizing the importance of staying hydrated. These resources contribute to practical recommendations for maintaining proper hydration and preventing dehydration-related issues during outdoor activities.

**Table 1.** Checklist for health concern locations, activities, and tools at the attractions.

| No. | Item | Yes | No | Free/Sale | Number of Points | Locations of The Points | Area Covered per Point | Remark |
|-----|------|-----|----|-----------|-------------------|--------------------------|------------------------|--------|
| 1 | Easy access drinking water | | | | | | | |
| 2 | Shade and Rest Areas | | | | | | | |
| 3 | Information and Education | | | | | | | |
| 4 | Proper Ventilation | | | | | | | |
| 5 | Emergency Response Plans | | | | | | | |
| 6 | Time Restrictions | | | | | | | |
| 7 | Monitoring and Surveillance | | | | | | | |
| 8 | Personalized Reminders | | | | | | | |
| 9 | Staff Training | | | | | | | |
| 10 | Accessibility to Medical Assistance | | | | | | | |
| 11 | Weather Monitoring | | | | | | | |

Ensuring tourist safety and protecting against heat-related illnesses necessitates the provision of accessible and complimentary drinking water points at tourist attractions. Alongside this, shaded rest areas such as pavilions, umbrellas, or natural tree cover should be made readily available. Cooling stations should also be implemented, including misting fans, air-conditioned zones, or shaded areas with fans. It is crucial for attraction managers to prioritize staff education and preparedness for emergencies, monitor weather conditions closely, and offer information on local health services. Particular emphasis should be placed on safeguarding vulnerable populations, including the elderly, children, pregnant women, and individuals with pre-existing health conditions, by offering customized accommodation and support. Maintaining and ensuring the cleanliness of facilities is of utmost importance to uphold the well-being and safety of visitors, especially in hot weather conditions. By implementing these measures, tourist attractions can enhance the safety and comfort of all visitors, fostering a secure and enjoyable experience.

### 3.2.2. Collection of Data from Internet Sources

Figure 3 shows the methodology used to obtain the data the tourist needed.

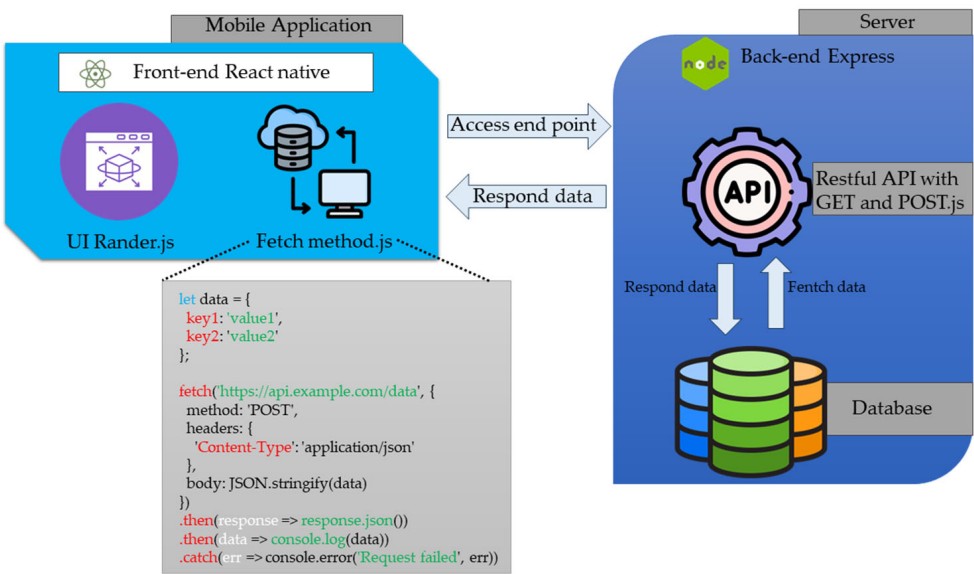

**Figure 3.** Framework of the method used to obtain data from internet sources.

Utilizing Figure 3 as a basis, our team developed a health tourism application using React Native and Node.js. The application consists of two distinct components: the Front-end React Native and the Back-end Express. The Front-end component focuses on UI rendering and data transmission to the server through fetch code. The fetch method encompasses

various parameters, including defining the API URL for server communication, specifying the sending method as either GET or POST and setting headers such as "Authorization" for data access rights obtained during user registration. On the other hand, the Back-end Express component receives data from the front-end fetch method and interacts with the database. It retrieves data in the case of a GET request and records or updates information in the case of a POST request. All data is transmitted in JSON format and can be seamlessly displayed on the application.

### 3.2.3. Collection of Data from Tourists via Applications and Smartwatches

The framework for obtaining data from tourists via mobile applications is shown in Figure 4.

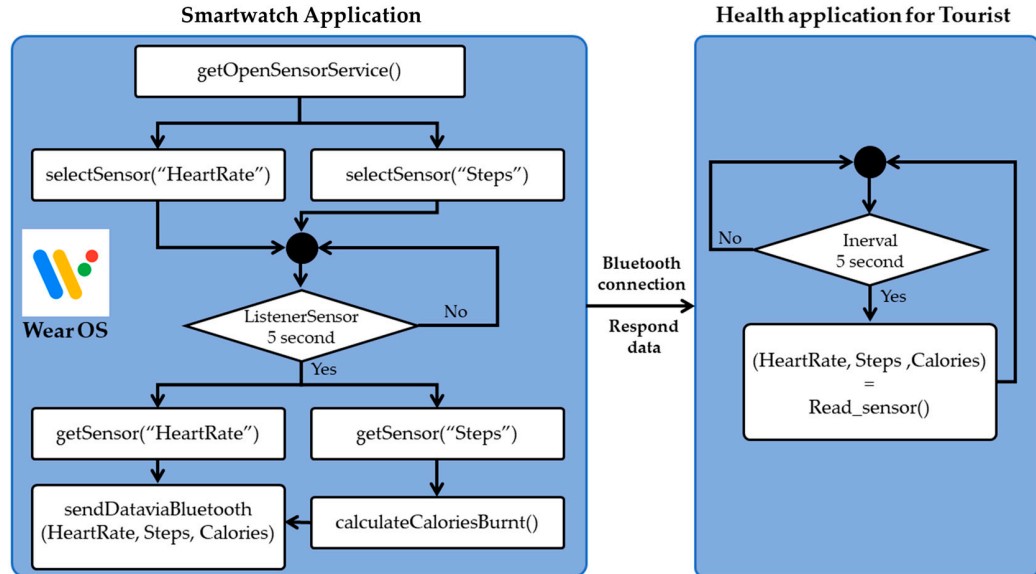

**Figure 4.** Framework of the method to obtain data from tourists via smartwatches.

In reference to Figure 4, we have developed an application compatible with smartwatches featuring Wear OS, an Android-based platform for application development. Before initiating the system, it is necessary for the smartwatch to establish a Bluetooth connection with a smartphone. To enable access to all sensors on the smartwatch, the getOpenSensorService is activated within the developed application. Specifically, we select the cardiac sensor (selectSensor("HeartRate")) and the step counter (selectSensor("Steps")). These sensors are configured to read data continuously every 5 s (ListenerSensor 5 s). Consequently, every 5 s, the application retrieves readings from the heart sensor (getSensor("HeartRate")) and step count (getSensor("Steps")). Subsequently, the pedometer sensor value is utilized to calculate calories burned (calculateCaloriesBurnt). This data, comprising heart rate, step count, and calorie count, is then transmitted via Bluetooth to the Health application for Tourist on the connected Smartphone. Within the Health application for Tourist, a function named Read_sensor is employed to retrieve sensor data from Bluetooth every 5 s.

### 3.2.4. Tourist Safety Support System Design

The design of the Tourist Safety Support System (T3S) involves the utilization of JavaScript as the programming language for the front-end and Node.js for the back-end. T3S has been developed with a user-friendly approach, ensuring ease of use for tourists. The framework of the T3S is illustrated in Figure 5, providing a visual representation of its structure and components.

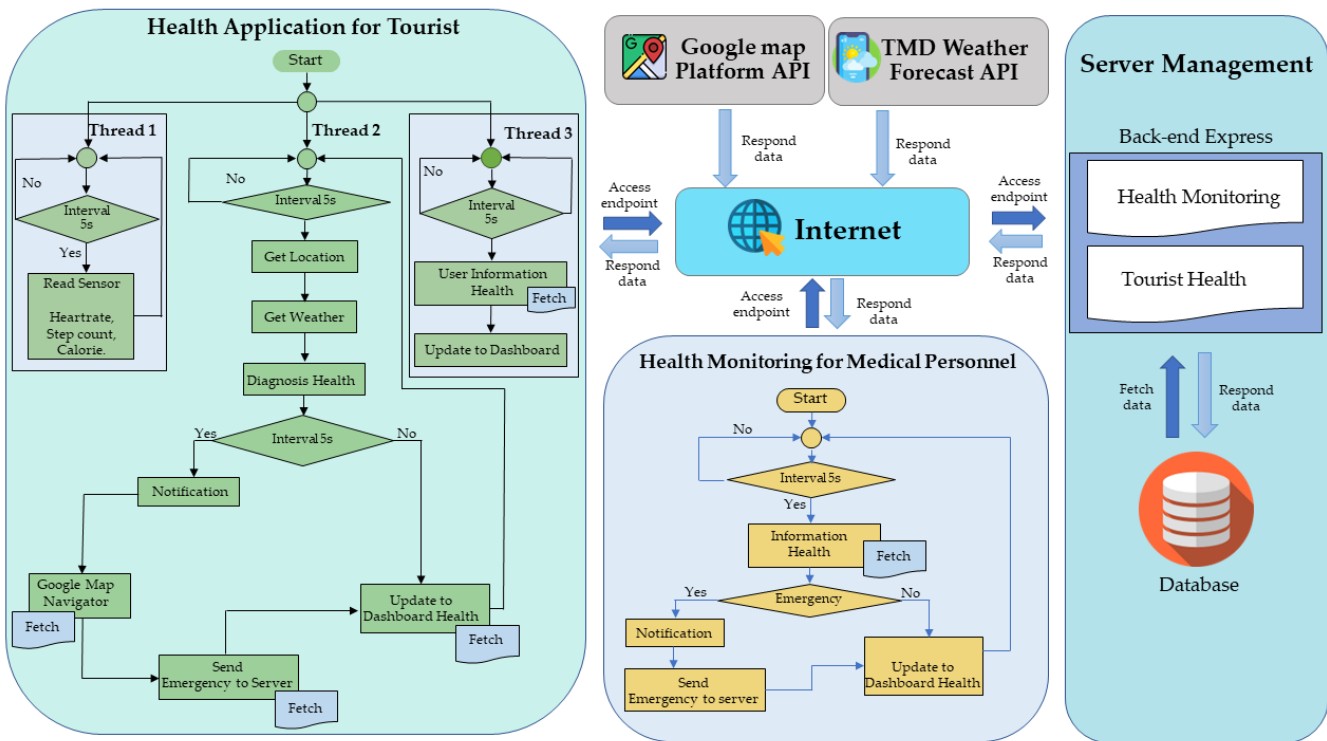

**Figure 5.** Framework of the TSSS (T3S).

Based on Figure 5, the tourist health application system is created utilizing Reazct Native, enabling cross-platform development for iOS and Android. It consists of three components: Tourist Health Application, Medical Staff Health Monitoring, and Server Management. The Tourist Health Application is divided into three threads. Thread 1 collects sensor data from smartwatches, and Thread 2 fetches weather information and assesses somatic conditions using heart rate and heat index data, notifying tourists of anomalies and facilitating emergency support. Thread 3 retrieves health data from the server's database and displays it on the application dashboard.

The Health Monitoring for Medical Staff application provides real-time tracking and monitoring of tourist health. It retrieves health information from the server's database, displays emergency alerts, and enables medical staff to access both tourist health data and coordinates. The Server Management component handles data transmission and storage, utilizing a Restful API architecture for seamless retrieval (GET) and database updates (POST) between the Health Application for Tourist and Health Monitoring for Medical Staff applications.

To evaluate the effectiveness of the designed tourist safety support system (T3S), a usability test was conducted involving 427 volunteers. The System Usability Scale (SUS) questionnaire, a well-established tool, was utilized and shown in Table 2. The SUS questionnaire consists of 10 items measured on a 5-point Likert scale, assessing aspects such as ease of use, learnability, efficiency, and satisfaction. Aggregated SUS scores range from 0 to 100, with higher scores indicating better usability. SUS is widely employed in both academia and industry to gather user feedback, inform design decisions, and create user-friendly systems. Its brevity and simplicity make it a popular choice for evaluating various software applications, websites, mobile apps, and hardware devices.

Besides the SUS, we wanted to determine how tourists liked our suggestion system. We also conducted a survey with the same tourists who discovered the SUS. Table 3 shows the questions.

**Table 2.** SUS Survey Prompts.

| No. | Questions | Strongly Disagree | | | Strongly Agree | |
|---|---|---|---|---|---|---|
| | | 0 | 1 | 2 | 3 | 4 |
| 1 | I would be motivated to use this system frequently. | | | | | |
| 2 | The system appeared unnecessarily intricate to me | | | | | |
| 3 | experienced the system as straightforward to use. | | | | | |
| 4 | I would seek technical guidance to utilize this system | | | | | |
| 5 | The various functions in this system were skillfully combined | | | | | |
| 6 | I witnessed inconsistencies in this system | | | | | |
| 7 | I trust that most people would quickly grasp the mechanics of operating this system. | | | | | |
| 8 | The system was unwieldy and arduous to navigate. | | | | | |
| 9 | I exuded a sense of assurance while navigating the system | | | | | |
| 10 | I recognized that I had to gain extensive understanding and proficiency in order to effectively operate this system. | | | | | |

**Table 3.** Exemplary Survey Instruments for Assessing Tourist Engagement and Loyalty.

| No. | Questions | 1 | 2 | 3 | 4 | 5 |
|---|---|---|---|---|---|---|
| 1 | What level of satisfaction of you with your overall experience at, taking into consideration the recommendations provided by the tourist recommendation system? | | | | | |
| 2 | How likely are you to recommend the attractions to others based on the recommendations provided by the tourist recommendation system? | | | | | |
| 3 | You use the recommendation system very often to decide which tourist attractions to visit during your trip? | | | | | |
| 4 | You follow every recommended attractions that recommend by the system | | | | | |
| 5 | The recommendation system is highly contribute to your engagement in the activities and attractions | | | | | |
| 6 | The recommendation system is highly influence your emotional connection with the attraction | | | | | |
| 7 | I will definitely use the recommendation system in the future. | | | | | |
| 8 | I will likely to share my positive experience with the tourist recommendation system and its impact on your visit to the attraction on social media or through word-of-mouth?- | | | | | |
| 9 | If the tourist recommendation system enhancing your engagement? | | | | | |

## 4. Result

### 4.1. Heat-Related Illnesses and Safety for Tourists: Key Weather Conditions and Information Elements to Consider When Visiting Hot and Humid Destinations

Based on the analysis of the literature and the insights provided by medical staff, the following weather conditions were identified as relevant factors in relation to heat-related illnesses and safety for tourists:

- Extreme heat
- Humidity
- UV levels

These weather conditions can be used to derive the following information given in Table 4.

Table 4 presents vital information elements concerning tourist safety and heat-related illnesses. Factors such as humidity, temperature, weather forecast, shaded areas, tourist temperature, average temperature increase, drinking water availability, total steps, and calories expended are crucial in empowering tourists to make informed decisions and prevent heat-related illnesses in hot and humid destinations. Weather conditions such as heat waves, extreme heat, and high humidity pose significant risks for heat-related illnesses, including heat stroke, dehydration, and cardiovascular events.

Taking precautionary measures such as staying hydrated, minimizing sun exposure, resting in shaded areas, wearing appropriate clothing, and adhering to heat safety guidelines greatly reduces the likelihood of these illnesses. Individuals with preexisting health

conditions should exercise extra caution. Information sources for each factor vary, with some obtained from the internet, others calculated through the mobile application, and specific data acquired from connected smartwatches. Primary data was also collected through surveys conducted at the destination. Further details regarding information sources can be found in Table 4.

**Table 4.** List of factors given to the tourists via the mobile application.

| No. | Factors | Source of Information |
|---|---|---|
| 1 | Level of humidity | Internet sources |
| 2 | Current temperature at the attraction | Internet sources |
| 3 | Current weather forecast at the destination | Internet sources |
| 4 | Percent of shaded and rest areas at the destination | Surveying of the destination |
| 5 | Current tourist temperature | Smartwatch/application |
| 6 | Current heart rate and pulse of the tourist | Smartwatch/application |
| 7 | Average temperature increase per 10 min | Smartwatch/application |
| 8 | Air quality index | Internet sources |
| 9 | Heat index | Internet sources/application calculation |
| 10 | Wind speed and direction | Internet sources |
| 11 | UV index | Internet sources |
| 12 | Availability and accessibility of medical facilities in the area | Surveying of the destination |
| 13 | Altitude of the destination | Internet sources |
| 14 | Number and location of drinking water services at the destination | Surveying of the destination |
| 15 | Total steps and calories expended at the destination | Smartwatch/application |

### 4.2. Case Study Detail

Ubon Ratchathani, situated in northeastern Thailand near the Thailand–Laos border, is an intriguing subject for academic research due to its remarkable history, cultural significance, and natural splendor. The province features ancient temples like Wat Nong Bua and Wat Thung Sri Muang, showcasing exceptional architectural design, mural paintings, and intricate wood carvings that reflect the region's cultural heritage. Ubon Ratchathani also offers pristine natural attractions, including Thung Si Mueang Park, Pha Taem National Park with prehistoric cave paintings, and Phu Chong Na Yoi National Park with majestic mountains, captivating waterfalls, and verdant forests. The proposed system incorporates the top 50 attractions with the highest average rating scores, as derived from the assessments of tourists on renowned platforms such as www.wongnai.com (accessed on 6 June 2023) and www.tripadvisor.com (accessed on 6 June 2023).

A highlight of Ubon Ratchathani is the annual Candle Festival held in July, featuring grand parades and meticulously crafted wax candles that showcase local craftsmanship and cultural traditions. The province is renowned for its traditional crafts, such as silk weaving and pottery, as well as vibrant local markets that provide an immersive experience of the local culture and cuisine. Overall, Ubon Ratchathani presents a unique combination of historical, cultural, and natural attractions, making it an enchanting destination for travelers seeking to explore the northeastern region of Thailand and immerse themselves in its rich heritage and natural beauty. Refer to Figure 6 for an illustration of Ubon Ratchathani's attractions.

Table 4 presents examples of the data obtained from observational research and verified using a comprehensive checklist. This data will be disseminated to tourists upon request through a mobile application. Table 5 shows the information on the tourist attraction obtained from observing the attractions, while Tables 6 and 7 show the information obtained from internet sources and the results obtained from the smartwatch and the designed application, respectively.

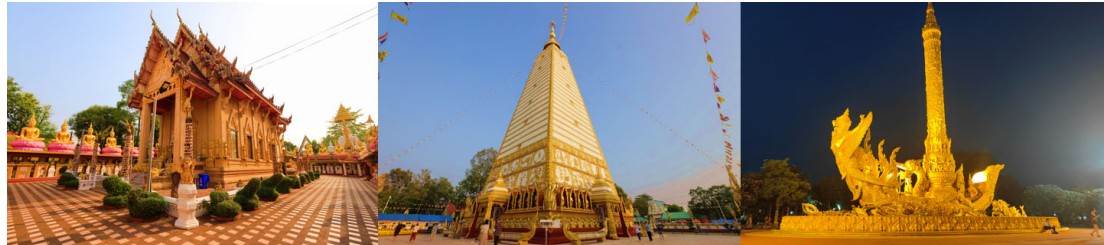

**Figure 6.** An example of the tourist attractions in Ubon Ratchathani.

**Table 5.** List of the factors that affect heat-related illness obtained from observing attractions.

| No. | Number of Drinking Water Service Points | % of Shade and Rest Area Compared with the Total Area of the Attraction | Information about the Attraction | Proper Ventilation | Emergency Response Plans | Time Restriction |
|---|---|---|---|---|---|---|
| UB1 | 2 | 23% | Yes | Yes | No | 9.00–18.00 |
| UB2 | 0 | 18% | No | Yes | No | 7.00–17.00 |
| UB3 | 0 | 90% | No | Yes | No | 8.00–20.00 |
| . . . | . . . | . . . | . . . | . . . | . . . | |
| UB14 | 1 | 70% | Yes | No | No | 9.00–18.00 |
| UB15 | 4 | 25% | No | Yes | No | 9.00–20.00 |

**Table 6.** List of factor values obtained from internet sources.

| No. | Current Temperature (Celsius) | UV Index | Altitude (Meters) | Humidity (RH %) | Wind Speed ((km/h) | Wind Direction (Degrees) | Heat Index (Celsius) | PM.2.5 (µg/m³) | Weather Forecasting |
|---|---|---|---|---|---|---|---|---|---|
| UB1 | 37 | 7 | 2565 | 64 | 12 | 80 | 41 | 67 | Sunny |
| UB2 | 36 | 8 | 2001 | 63 | 13 | 0 | 42 | 80 | Slight rain (20%) |
| UB3 | 36 | 12 | 2045 | 61 | 15 | 70 | 39 | 56 | Sunny |
| . . . | . . . | . . . | . . . | . . . | . . . | . . . | . . . | . . . | |
| UB14 | 34 | 9 | 546 | 78 | 15 | 80 | 38 | 110 | Sunny |
| UB15 | 38 | 10 | 798 | 60 | 13 | 90 | 40 | 82 | Possible rain (10%) |

**Table 7.** List of factor values obtained from the smartwatch and designed application (example from 5 tourists).

| Tourist Number | Current Tourist Temperature (Celsius) | Current Heart Rate and Pulse of the Tourist (bpm) | Average Temperature Increase per 10 min (Celsius) | Total Steps and Calories Expended at the Destination (Steps/Calories) |
|---|---|---|---|---|
| 1 | 36.8 | 89 | 0.03 | 5600/320 |
| 2 | 36.1 | 78 | 0.04 | 5100/210 |
| 3 | 37.0 | 91 | 0.00 | 6700/294 |
| 4 | 35.9 | 84 | −0.10 | 12,123/676 |
| 5 | 36.5 | 92 | 0.04 | 14,818/760 |

Walker and Dallas [62] suggested that body temperature can vary from 0.28–0.35 Celsius. The data presented in Tables 4–6 correspond to the information provided in Table 4, which is crucial for informing tourists about protecting themselves from heat-related illnesses. The use of all the information in Tables 4–6 for designing the T3S and enhancing the safety of tourist travel is discussed in Section 4.3.

### 4.3. The T3S Effectiveness Testing

The effectiveness of the T3S has been divided into three parts. These are (1) test for the accuracy of the T3S in obtaining and providing data, (2) test for user satisfaction subjected to the designed application, and (3) test effectiveness to promote attractions loyalty and engagement. The application can be downloaded at https://play.google.com/store/apps/details?id=com.aiotour (accessed on 6 June 2023)

#### 4.3.1. Test for the Accuracy of the T3S in Obtaining and Providing Data

The T3S system has been meticulously designed to deliver comprehensive information about tourist attractions, particularly concerning health safety factors that may contribute to heat stroke, dehydration, and heart attacks. These factors encompass temperature, humidity level, atmospheric conditions, pulse rate, heart rate, body temperature, and blood pressure.

Based on the data presented in Table 8, the accuracy percentage signifies the extent to which the application aligns with the actual weather conditions, with a perfect match denoted by 100%. Meanwhile, the response time, measured in seconds, represents the duration it takes for the application to retrieve and process the information upon activation. For instance, the application demonstrates 100% accuracy in reporting the current temperature, indicating its correspondence with the real-time temperature data. Additionally, the response time for temperature retrieval and processing stands at 0.002 s, reflecting the efficiency of the application in accessing and analyzing information from internet sources.

**Table 8.** The accuracy of the current weather condition.

|  | Accuracy | Response Time (s) | Precision |
|---|---|---|---|
| Temperature | 100 | 0.002 | 100 |
| Humidity | 100 | 0.001 | 100 |
| Weather condition | 100 | 0.000 | 100 |
| Pulse | 100 | 0.000 | 100 |
| Heart rate | 100 | 0.000 | 100 |
| Body temperature | 100 | 0.001 | 100 |
| Blood pressure | 100 | 0.001 | 100 |
| Chance of heat stroke | 100 | 0.000 | 100 |
| Chances of dehydrating | 100 | 0.000 | 100 |
| Average | 100 | 0.001 | 100 |

Similarly, the application demonstrates 100% accuracy in reporting humidity and weather conditions, with response times of 0.001 s and 0.000 s, respectively. This guarantees the provision of swift and precise information about current weather conditions to users. The average accuracy and response time for all weather parameters are 100% and 0.001 s, respectively. Consequently, the application ensures the reliable delivery of up-to-date weather information to users in a timely manner.

#### 4.3.2. Test for User Satisfaction Subjected to the Designed Application

The study involved 427 respondents who provided their feedback on the usability of the mobile application for tourism safety. The methodology used for data collection is detailed in Section 3.2.4. The average System Usability Scale (SUS) score obtained from the respondents was 97.08, indicating a high level of usability and a well-thought-out design for the developed software. These results strongly suggest that the mobile application tailored for tourism safety is user-friendly, providing a positive user experience. The consistent and favorable average SUS score further reinforces the notion that the application is reliable and efficient in meeting users' needs.

Upon closer examination of the individual responses, it was observed that none of the 427 respondents rated the application below the SUS score of 80. A significant

number of users, 180 individuals to be precise, awarded the application a full SUS score, demonstrating a high level of satisfaction with the developed software. This finding is particularly noteworthy as it indicates that 42.16% of all respondents not only preferred using the application themselves but also expressed a willingness to recommend it to others. Additionally, the study assessed the average scores of all 10 questions presented in Table 2. The question that received the lowest rating from respondents was "I would be motivated to use this system frequently", with a score of 93.21 out of 100 (Figure 7). The possible reason for this lower rating could be attributed to respondents' limited exposure to hot and humid areas for tourist attractions, leading to less frequent usage of the application. Despite this, the overall satisfaction level of the respondents remained high.

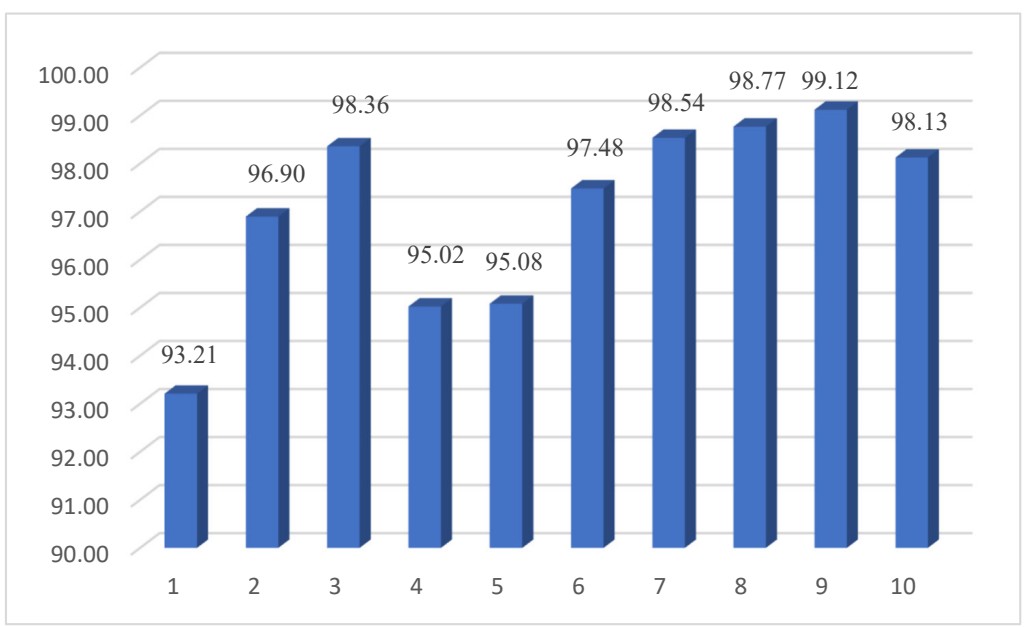

**Figure 7.** The average score of each question.

On the other hand, the question that received the highest score was "I exuded a sense of assurance while navigating the system", obtaining a satisfaction level of 99.12. This result clearly indicates that the program's ease of use instilled confidence in the respondents, making them feel comfortable while using the application. Considering the compelling data and positive feedback received from the respondents, it is evident that the developed mobile application for tourism safety is indeed an impressive and reliable solution. The high SUS scores and user satisfaction levels endorse the application's effectiveness in catering to users' needs and preferences. As an informed opinion, the results of this study inspire confidence in the usability and design of the mobile application. The substantial percentage of respondents expressing a willingness to recommend the application further strengthens the validity of the findings. It is evident that the software has been well-received by the target audience, indicating its potential to positively impact the tourism safety domain.

The T3S system is specifically developed to assist tourists in traveling safely by monitoring their body temperature and offering relevant safety recommendations and precautions. Consisting of a mobile application and a wearable device, which can be conveniently worn on the wrist, neck, or waist, the system effectively measures the user's body temperature and transmits the collected data to the application for thorough analysis. Based on the analyzed data, the app provides personalized safety recommendations and precautions to the user. Furthermore, the T3S application provides users with up-to-date information regarding their real-time location, prevailing weather conditions, and other relevant factors, thereby augmenting overall safety and comfort levels.

The application provides a comprehensive health monitoring system for travelers, as showcased in Figure 8a. The system captures basic health information during the

registration process, including weight, height, BMI, gender, age, and blood type, which enables the application to predict potential health risks. Additionally, GPS coordinates on the traveler's smart device are used to retrieve weather information from their location. The application calculates the heat index and displays this information on the screen to alert travelers to potentially harmful weather conditions. The application incorporates sensor data from smartwatch devices, such as heart rate, steps taken, and calories used, to predict and diagnose the traveler's health status. The application identifies six different health statuses and alerts users to harmful heat index zones to ensure their safety while traveling.

Figure 8b showcases the application's emergency medical assistance feature, which displays a pop-up alert to inquire about a traveler's abnormal symptoms. If there is no response, the system automatically sends a request for medical assistance to the nearest facility, ensuring prompt attention during emergencies. The application's search capabilities for tourist attractions, restaurants, hospitals, hotels, and stores are displayed in Figure 8c. The search function allows users to search by location name or type, making it easy to navigate and find the desired location. Search results display a preview of the place, including the place name, address, and distance from the current location coordinates.

Figure 8d provides detailed information about a specific location, including a gallery of up to eight preview photos, coordinates for system navigation, location type, telephone number, address, and nearby grocery stores. Additionally, the application provides a daily weather forecast for the next six hours, including temperature, humidity, and visual signals or icons to quickly inform users of the weather conditions. The application calculates and displays heat index zones to inform users of potential health risks due to weather conditions. Figure 8e showcases the map screen displaying the location of tourist attractions, which works in conjunction with the Tourist Location and Health Information screens to provide location coordinate data. The application generates a marker on Google Maps for each location, enabling users to navigate to their desired destination by clicking on the marker.

### 4.3.3. Test Effectiveness to Promote the Attraction's Loyalty

Based on the findings presented in Table 9, this study aimed to assess tourist loyalty and engagement toward a tourist recommendation system. The sample consisted of 427 tourists, categorized into six groups based on gender and age. The overall results revealed a high average score ranging from 4.58 to 4.92 across all nine questions, indicating general satisfaction with the tourist recommendation system.

Notably, the sixth category, representing tourists over the age of 45, exhibited the highest average score of 4.87, indicating their heightened satisfaction with the recommendation system. This result suggests the system's effectiveness for older tourists who may encounter challenges when navigating unfamiliar destinations and activities. Moreover, the findings indicated variations in satisfaction levels among different age and gender groups. Specifically, the second age group (over 45 years) attained the highest average score (4.81) among all age groups, indicating their elevated engagement and loyalty toward the recommendation system. Conversely, the first age group (18–30 years) had the lowest average score (4.71), suggesting relatively low satisfaction with the system. These outcomes highlight the importance of tailoring the recommendation system to address the specific needs and preferences of diverse age groups.

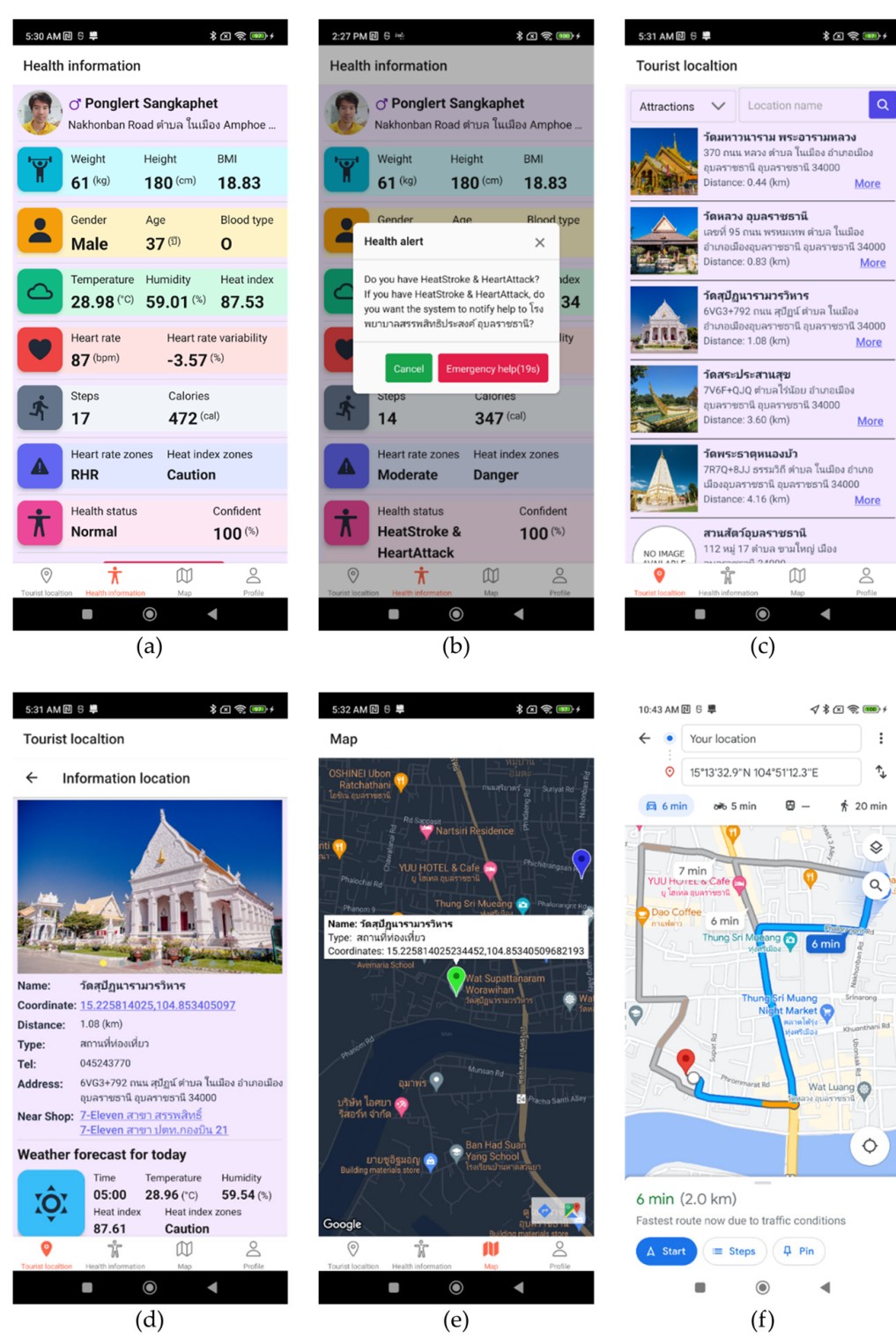

**Figure 8.** Dashboard screen of tourist health information (**a**) basic health information report, (**b**) abnormal health warning, (**c**) main page of pre–caution attraction's information of the attraction, (**d**) detail page of pre–caution attraction's information, (**e**) attraction's location page and (**f**) traveling route to the target location.

**Table 9.** Results of the engagement and loyalty questionnaire.

| Age of Respondents | 18–30 years | | 31–45 years | | ≥45 years | |
|---|---|---|---|---|---|---|
| Number of respondents | 171 | | 164 | | 182 | |
| % of respondents | 33.08 | | 31.72 | | 35.20 | |
| Gender | Male | Female | Male | Female | Male | Female |
| Number of respondents (gender) | 90 | 81 | 84 | 80 | 75 | 107 |
| % of respondents (gender) | 52.63 | 47.37 | 51.22 | 48.78 | 41.21 | 58.79 |
| What level of satisfaction of you with your overall experience at, taking into consideration the recommendations provided by the tourist recommendation system? | 4.58 | 4.81 | 4.51 | 4.73 | 4.74 | 4.79 |
| How likely are you to recommend the attractions to others based on the recommendations provided by the tourist recommendation system? | 4.64 | 4.70 | 4.67 | 4.71 | 4.77 | 4.83 |
| You use the tourist safety recommendation system very often to decide which tourist attractions to visit during your trip? | 4.56 | 4.68 | 4.61 | 4.82 | 4.64 | 4.92 |
| You follow every recommended attractions that recommend by the system | 4.61 | 4.69 | 4.67 | 4.79 | 4.72 | 4.88 |
| The recommendation system is highly contribute to your engagement in the activities and attractions | 4.76 | 4.81 | 4.71 | 4.81 | 4.78 | 4.91 |
| The recommendation system is highly influence your emotional connection with the attraction | 4.72 | 4.77 | 4.68 | 4.71 | 4.84 | 4.87 |
| I will definitely use the recommendation system in the future. | 4.71 | 4.69 | 4.64 | 4.74 | 4.71 | 4.80 |
| I will likely to share my positive experience with the tourist recommendation system and its impact on your visit to the attraction on social media or through word-of-mouth? | 4.74 | 4.81 | 4.71 | 4.82 | 4.84 | 4.88 |
| If the tourist recommendation system enhancing your engagement? | 4.70 | 4.78 | 4.69 | 4.81 | 4.76 | 4.91 |
| Average | 4.67 | 4.75 | 4.65 | 4.77 | 4.76 | 4.87 |
| % Average | 93.38 | 94.98 | 93.09 | 95.42 | 95.11 | 97.31 |

## 5. Discussion

In this section, we present the study's findings and implications of implementing a real-time information system for tourists at destinations. The system provides data on temperature, humidity, weather conditions, and safety tool preparation. The implications of this system are wide-ranging, encompassing tourism management, policy effectiveness, impact on tourist behavior, destination management, and practical implications. Additionally, the system contributes to sustainable tourism practices and opens up potential future directions.

### 5.1. The Impact of Weather and Safety Information on Tourist Safety and Risk Mitigation

The provision of weather and safety information has become a vital determinant of tourist behavior. Our study demonstrates that tourists equipped with such information exhibit a higher propensity to actively prepare for adverse weather conditions, such as extreme temperatures or inclement weather, by acquiring suitable attire, equipment, and safety tools. This signifies the system's potential to considerably enhance tourist safety and mitigate the risks associated with weather-related accidents or injuries. Additionally, our findings indicate that tourists display greater adherence to safety guidelines and recommended safety tools, underscoring the positive influence of the system in promoting improved safety practices among tourists. These heightened safety measures can contribute

to increased tourist satisfaction, subsequently influencing their loyalty and engagement with attractions.

These findings align with previous research by Jeuring and Peters [63] and Lohmann and Hübner [64], which support the idea that providing tourists with pre-trip weather information positively impacts their travel experiences. The study cited in Jeuring and Peters [63] suggests that access to weather information enables tourists to effectively plan and prepare for their trips, resulting in heightened enjoyment and overall satisfaction. Furthermore, the study mentioned by Lohmann and Hübner [64] indicates that tourists who have access to weather information prior to their trips are more inclined to confidently engage in outdoor activities, such as hiking and sightseeing, as they possess better awareness of prevailing weather conditions and can make appropriate choices regarding attire and activities.

The results obtained from the evaluation of our environmental monitoring system (Table 8) underscore its outstanding performance across all key metrics. Notably, the system achieved a perfect score of 100% in accuracy, indicating its ability to deliver precise and reliable readings without any errors or deviations. This level of accuracy is crucial for environmental monitoring, where even minor inaccuracies could lead to erroneous conclusions and misinformed decision-making. Equally impressive are the system's response times, ranging from 0.000 to 0.002 s, which sets a new standard for swift data processing. Such near-instantaneous response times are of paramount importance in environmental monitoring, as timely data acquisition is vital for understanding rapidly changing environmental conditions and detecting potential hazards promptly.

The system's precision, also achieving a perfect score of 100%, showcases its consistency in producing identical results for repeated measurements. This level of precision fosters confidence in the system's reliability, particularly in critical applications where consistent and dependable data is essential for drawing accurate conclusions. It is evident from the comprehensive dataset that the system performs exceptionally well across diverse environmental factors. The ability to monitor temperature, humidity, weather conditions, vital signs, and health risks (heat stroke and dehydration) with such high accuracy and precision render this system an invaluable asset for various sectors, including public health, agriculture, and disaster management.

*5.2. Weather and Safety Information System: Implications for Tourist Safety and Destination Management*

The system's implications for destination management align with existing policies and regulations prioritizing tourist safety. Providing weather and safety information enhances destination preparedness for weather challenges, enabling timely warnings and recommendations. This proactive approach prevents negative impacts on health, safety, and destination reputation. Successful implementation requires collaboration among stakeholders, including tourism authorities, weather agencies, and local businesses. Sharing data and resources ensures accurate and timely information. Costs, technology, standardization, and regulation of information provision should be considered. Agarwal et al. [65] support these recommendations. The system promotes responsible tourism by empowering informed decisions and precautions, reducing risks, and enhancing destination sustainability, reputation, and visitor satisfaction.

Mobile applications for tourism safety have been proposed in several studies. These applications aim to address concerns about criminal incidents and safety issues during travel. The applications provide features such as preventive measures, first-aid assistance, guidance tracking, and safety information. They utilize technologies like beacons Bluetooth Low Energy (BLE), mobile device apps, and cloud servers. The applications also incorporate features like context information, collaborative gamified approaches, and real-time emergency information. The proposed applications offer various functions such as tourism data collection, advice, orientation for ecological trails, and tourism queries. They leverage the capabilities of smartphones, including sensors, GPS navigation, and high-speed internet

connectivity. These applications contribute to enhancing the safety and security of tourists during their journeys.

Researchers [44] have successfully developed an innovative interactive design for a tourism app that places the utmost emphasis on enhancing safety measures across diverse tourist destinations. Their approach encompasses both preventive strategies and timely first-aid assistance, effectively addressing potential criminal incidents. Additionally, Akter et al. [45] introduced a mobile application primarily dedicated to ensuring traveler safety during road trips. This application offers comprehensive safety provisions and prompt aid throughout the journey, thereby providing a worry-free travel experience.

Upon comparing these two articles, it becomes evident that the proposed application boasts superior functionalities compared to its predecessors. Notably, the proposed application includes a risk warning feature that takes into account weather conditions, a facet that previous research has not addressed. Furthermore, the application is thoughtfully designed to connect its warning system with the nearest medical centers, enabling swift response to any unforeseen conditions tourists may encounter. This real-time monitoring of tourists' well-being during site visits is an exceptional characteristic unique to the proposed application, setting it apart from other existing solutions.

Moreover, the proposed application offers an additional feature of real-time updates on heat-related illness risks, ensuring that tourists receive up-to-date information to safeguard their well-being. This proactive approach in providing crucial safety-related information is a testament to the application's dedication to enhancing tourist safety and further highlights its superiority over other alternatives.

### 5.3. The Relationship of the SUS Score, Tourist's Satisfaction, Loyalty, and Engagement

Our study indicates that the implemented system delivers accurate and up-to-date information to tourists with minimal delay (0.001 s). The provided weather and safety information is perceived as reliable and timely, enabling effective preparation and informed decision-making. Users find the system accessible, comprehensible, and user-friendly. The System Usability Scale (SUS) was administered to 427 tourists, resulting in an overall score of 97.45, signifying high satisfaction [66]. A higher SUS score correlates with customer satisfaction, impacting loyalty and engagement [67,68]. Ease of use, efficiency, and effectiveness foster positive experiences, loyalty, and engagement [69]. A high SUS score increases customer engagement, while a low score diminishes loyalty and engagement. Note that other factors influence the relationship between SUS score, loyalty, and engagement. Therefore, interpreting SUS scores alongside relevant data and considering system features and context is crucial.

This study examines the association between SUS score and tourist loyalty within a proposed application. Findings show a significant positive relationship ($p < 0.001$), supporting Dianat et al.'s [70] idea on the relationship between SUS score and customer satisfaction. Statistical analysis details, including the SUS score and tourist loyalty relationship, can be found in Table 10.

**Table 10.** Information from the regression analysis revealed the relation between the SUS score and tourist loyalty.

| Indicator | R Square | Sy.x | F | DFn, DFd | *p*-Value | Equation |
|---|---|---|---|---|---|---|
| Value | 0.7614 | 1.476 | 89.35 | 1.28 | <0.001 | Y = 1.075 × X − 9.881 |

### 5.4. Gender and Age Effects on Tourist Loyalty: Insights into Tourism Safety Management and Trip Recommendation Systems

Table 11 presents the statistical test results conducted on the data obtained from Table 9. The objective was to assess whether respondents' gender and age range have a significant impact on the average satisfaction score for the developed software. The analysis of variance (ANOVA) was performed using a factorial design with a significance level of 0.05, using Minitab as the statistical tool. The ANOVA results in Table 11 indicate

that both the age range and gender of the respondents have a statistically significant effect on the average satisfaction score. Furthermore, there is a statistically significant interaction between the respondents' age range and gender, demonstrating that different age groups exhibit varying levels of satisfaction with the developed software. Specifically, female respondents, on average, provided higher satisfaction scores than male respondents. However, when examining the raw data of individual average satisfaction scores, both male and female respondents across all age ranges expressed the highest level of satisfaction (over 4.50) for the usability of the mobile application for tourism safety.

**Table 11.** Statistical test of the data from Table 9.

| Sources | DF | Seq.SS | Adj.SS | Adj.MS | F | $p$ |
|---|---|---|---|---|---|---|
| Age range | 2 | 1.65790 | 1.33192 | 0.66596 | 297.78 | 0.000 |
| Gender | 1 | 1.30402 | 1.30693 | 1.30693 | 584.38 | 0.000 |
| Age range*gender | 2 | 0.03736 | 0.03736 | 0.01868 | 8.35 | 0.000 |
| Error | 511 | 1.14281 | 1.14281 | 0.00224 | | |
| Total | 516 | 4.14209 | | | | |

This study offers valuable perspectives on tourist loyalty in the context of tourism safety management and the utilization of trip recommendation systems, specifically focusing on gender and age dynamics (refer to Figure 9). Notably, females exhibited a higher level of loyalty compared to males within the corresponding age bracket, showcasing a statistically significant difference of 2.18%. Furthermore, female tourists consistently exhibited greater loyalty across various age groups when utilizing the trip safety recommendation system. These findings have important implications for tourism practitioners and policymakers, highlighting the need for targeted strategies catering to the specific needs and preferences of diverse tourist segments.

Analysis of tourist loyalty across different age groups revealed interesting trends. Younger tourists (18–30 years old) exhibited comparatively lower loyalty, indicating potential differences in preferences and behaviors regarding tourism safety management and trip recommendation systems. Factors such as risk perception, travel experience, and decision-making processes may influence this disparity. Conversely, tourists aged over 45 displayed the highest level of loyalty, regardless of gender, suggesting their prioritization of safety in the travel decision-making process and inclination to utilize tourism safety management and trip recommendation systems for a secure and enjoyable travel experience.

Gender disparities in loyalty towards tourism safety management and trip recommendation systems were evident, showcasing varying degrees of loyalty between males and females across different age groups. Male tourists aged 18–30, 31–45, and over 45 years displayed loyalty scores of 4.67, 4.65, and 4.76, respectively. In contrast, their female counterparts in the same age groups exhibited slightly higher loyalty scores of 4.75, 4.77, and 4.87. These findings suggest that females might be more inclined to utilize such systems, regardless of age.

The interpretation of the results reveals that gender and age play significant roles in understanding tourist behavior and preferences towards tourism safety management and trip recommendation systems. The fact that female tourists showed higher loyalty scores in comparison to males indicates that safety and security are crucial factors for female travelers, potentially influencing their choice of destinations and activities. These findings highlight the importance for tourism businesses and managers to consider gender-specific preferences when designing safety measures and recommendation systems to effectively cater to their diverse customer base. By acknowledging these disparities, the industry can tailor its services better and enhance the overall experience for all tourists.

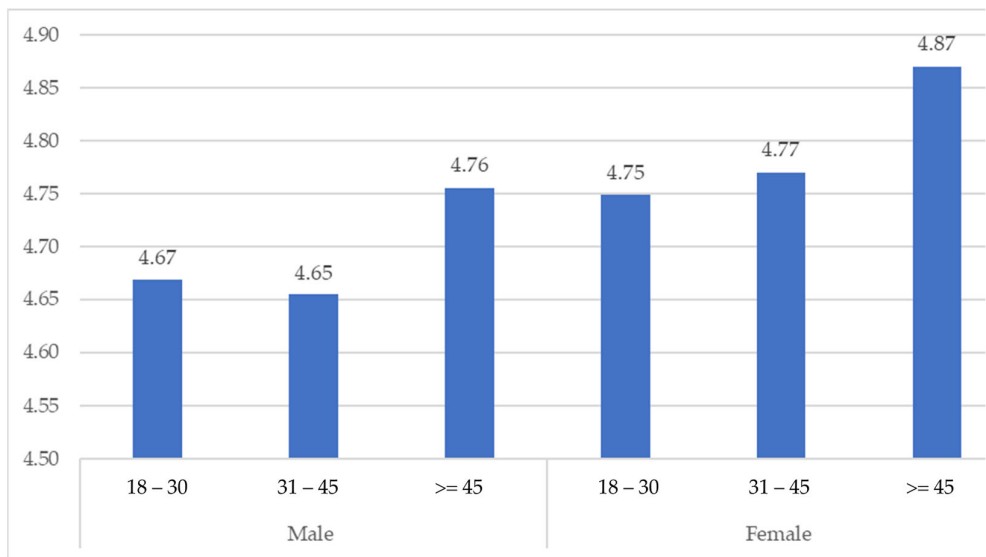

**Figure 9.** The average loyalty score of the various tourist types.

*5.5. Limitations and Potential Biases*

Several limitations and potential biases should be considered when interpreting the findings of this study. First, the sample size of our study was relatively small, which may limit the generalizability of our findings to other populations or contexts. While we attempted to recruit a diverse sample of tourists, our recruitment methods may have introduced selection bias, as we relied on convenience sampling and may have missed certain segments of the tourist population. Second, our study relied on self-reported data from tourists, which may be subject to social desirability bias or other forms of response bias. While we attempted to minimize these biases by ensuring anonymity and confidentiality, it is possible that some participants may have provided inaccurate or incomplete information.

Third, the interpretation of our data may be subject to researcher bias, as our team members may have preconceived notions or expectations about the effectiveness of the Tourist Safety Support System (T3S). To minimize this bias, we employed a rigorous data analysis process that involved multiple team members and peer review. Fourth, our study did not include objective measures of tourist safety and security, such as crime statistics or accident reports, which may limit the validity of our findings. While we attempted to assess tourist perceptions of safety and security through surveys and interviews, it is possible that these perceptions may not accurately reflect the actual safety and security conditions at the tourist attractions. Fifth, our study focused on a specific geographic region and tourist population, which may limit the generalizability of our findings to other regions or populations. While we attempted to recruit a diverse sample of tourists, it is possible that our findings may not be applicable to other contexts.

Finally, our study was conducted over a relatively short period of time, which may limit the ability to detect long-term effects or changes in tourist behavior. While we attempted to assess both short-term and long-term effects of the T3S, it is possible that longer-term effects may emerge over time. Despite these limitations and potential biases, we believe that our study provides valuable insights into the effectiveness of the Tourist Safety Support System (T3S) and its potential impact on tourist safety and security. We recommend that future research address these limitations and potential biases by employing larger sample sizes, more diverse recruitment methods, and more objective measures of tourist safety and security.

## 6. Conclusions and Outlooks

Our research introduces a mobile application aimed at augmenting tourist safety in hot and humid destinations. This innovative solution incorporates a range of advanced

features, including a comprehensive warning system, health recommendations, and a life rescue system. The findings of our study demonstrate the remarkable effectiveness of the implemented system, as it consistently delivers precise and timely weather and safety information to tourists, boasting an impressive average accuracy rate of 98.667% with an astounding response time of a mere 0.001 s.

This proposed application serves as an invaluable tool for destination service providers, enabling them to proactively address the diverse needs of tourists and ensure their utmost safety. Moreover, our study evaluated the impact of a tourist recommendation system on tourist loyalty and engagement, and the results were compelling. The system was found to be highly effective in enhancing tourist engagement and fostering loyalty towards tourist attractions. The consistently high satisfaction levels among tourists of different age and gender groups suggest that the system possesses a broad appeal and has the potential to significantly enhance the overall tourist experience.

Moving forward, future research pertaining to real-time weather and safety information systems for tourists should encompass a comprehensive assessment of long-term effectiveness. This research should also explore the implications for destination management, investigate technological advancements, and shed light on barriers to adoption. By undertaking such endeavors, we can collectively advance our understanding of tourism management and policy and further bolster the development and implementation of highly effective weather and safety information systems for tourists.

**Author Contributions:** Conceptualization, S.D., R.P. and N.N.; methodology, R.P. and P.S.; software, T.S., N.N. and P.S.; validation, S.K. and C.B.; formal analysis, S.D., T.S. and G.J.; investigation, C.B.; resources, T.S. and S.K.; data curation, S.D., G.J. and N.N.; writing—original draft preparation, N.N. and R.P.; writing—review and editing, C.B. and N.N.; visualization, T.S. and S.K.; supervision, R.P.; project administration, R.P. and N.N.; funding acquisition, T.S. All authors have read and agreed to the published version of the manuscript.

**Funding:** This research was funded by Thailand Science Research and Innovation (TSRI) and National Science, Research and Innovation Fund (NSRF). The grant number is 4110679.

**Institutional Review Board Statement:** Not applicable.

**Informed Consent Statement:** Not applicable.

**Data Availability Statement:** The data presented in this study are available on request from the corresponding author.

**Acknowledgments:** This research was supported by the Ministry of Higher Education, Science, Research, and Innovation (MHESRI), the Basic Research Fund group, and the AIO-SMART laboratory at Ubon Ratchathani University.

**Conflicts of Interest:** The authors declare no conflict of interest.

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
