# Peer review of "A Mobile Solution for Enhancing Tourist Safety in Warm and Humid Destinations"

_applsci, doi:10.3390/app13159027_

Round 1
Reviewer 1 Report
It is evident that this study offers a valuable contribution by proposing a mobile solution to enhance tourist safety in warm and humid destinations. The proposed solution encompasses a comprehensive warning system, health recommendations, and a life rescue system, all integrated into a mobile application. The effectiveness of the implemented system is demonstrated through its consistent delivery of precise and timely weather and safety information to tourists.
However, there are some important points that need to be addressed and supplemented to further strengthen the paper:
1. Derivation of research objective: The paper presents the research objective but fails to mention how it was derived. It is necessary to provide a clear explanation of the process and considerations that led to the formulation of the research objective.
2. Research questions based on existing studies: The research questions that underpin the research objective should be explicitly presented. These research questions should be developed based on the existing research hypothesis or research problem identified in previous studies.
3. Clearer justification in the introduction: The introduction should provide a clearer justification for the limitations of existing tourist information and how the proposed mobile application overcomes these limitations. Additionally, it would be beneficial to include a concise statement of the research hypothesis or research questions that will be addressed in the study, along with a more explicit statement of the significance and potential contributions of the proposed research to the field of tourism and tourist safety. It may also be useful to briefly mention any potential limitations or challenges that may arise in developing and implementing the proposed mobile application.
4. Additional references from "Applied Sciences" journal: The list of references should be expanded to include 3-5 articles from the "Applied Sciences" journal to enhance the literature review and support the study's claims and findings.
5. Potential problems and improvements in research methods:
a. Diversify data sources: To ensure a comprehensive understanding of tourist safety and preferences, it is recommended to include a wider range of data sources such as surveys, direct observations, and objective measurements.
b. Increase sample size and diversity: To improve the generalizability of the findings, it is essential to include a larger and more diverse sample of participants, encompassing tourists from different demographics and geographic locations.
c. Provide detailed data collection procedures: To enhance transparency and replicability, the paper should provide a thorough description of the data collection procedures, including observation protocols, selection criteria for attractions, and the process for validating findings from reference sources and interviews.
d. Conduct statistical analysis: Applying appropriate statistical analysis techniques to the collected data will enable the identification of significant findings, correlations, and meaningful conclusions.
e. Employ a comprehensive evaluation approach: Designing a robust evaluation plan that includes objective performance metrics, user feedback surveys, and comparative analysis with existing systems or benchmarks will help assess the efficacy of the proposed Tourist Safety Support System (T3S).
f. Discuss limitations and potential biases: Explicitly addressing the limitations of the research methods, potential biases in data collection or analysis, and other influencing factors will provide a more nuanced interpretation of the findings and guide future research.
6. Problems and improvements in the "Discussion and Conclusion" section:
1) Problems:
a. Lack of specific references: The discussion should include specific details and complete citations for the studies referenced, rather than making general references to previous research.
b. Incomplete comparison with previous studies: A more comprehensive comparison should be provided, outlining the specific similarities and differences between this study's findings and those of previous studies.
c. Lack of data interpretation: The discussion should delve into the underlying data and statistical analysis, providing more detailed interpretations and discussing the statistical significance of the relationship between SUS scores and tourist loyalty.
2) Improvements:
a. Provide specific references: Including specific references with complete citations will enhance the credibility and transparency of the discussion.
b. Detailed comparison with previous studies: A comprehensive comparison should be made, highlighting the similarities, differences, and unique contributions of this study compared to previous research.
c. Provide more data interpretation: Offering detailed data interpretation, including actual data points, statistical tests used, and measures of effect size or significance, will strengthen the conclusions drawn in the discussion.
By addressing these issues and incorporating the suggested improvements, the paper will demonstrate a higher level of polish and sophistication, enhancing the overall quality of the English sentences and the research itself.
The English sentences in the paper appear to be well-written and grammatically correct. However, the English sentences in this paper appear to have some issues and could benefit from improvement. Here are a few observations:
1. Sentence structure and clarity: Some sentences are lengthy and complex, making it challenging to follow the intended meaning. Breaking down these sentences into smaller, clearer sentences would enhance readability.
2. Lack of parallelism: In some instances, parallelism is lacking, leading to inconsistencies in sentence structure. Ensuring parallelism in lists, comparisons, and descriptions can improve the overall flow and coherence of the text.
Author Response
Thank you for your valuable comments and suggestion. We have try our best to resolve every issues. Please see the attached for the answers point-by-point.

Reviewer 2 Report
The submitted paper is devoted to an important challenge of using mobile applications specifically designed to enhance tourist safety in warm and humid destinations. The abstract fully highlights a main aspect considered in the presented research.
After manuscript reading, the next suggestions can be made:
1) Line 67. In my opinion, a better title for the second paragraph is “Literature Review”. Please, check other same issues according to the template.;
2) Figure 3. Please, provide better resolution;
3) Figure 7 shows that all respondents are nearly fully satisfied with using the proposed application. The results are great but questionable. It should be explained why, even on a small sample (30 random tourists), such a high level of satisfaction from using the application has been achieved. Please, add an explanation to the manuscript. It needs to be described.;
Finally, the research results have a sufficient practice aspect for safety tourism.
Please, check grammar and wording
Author Response
Thank you for your valuable feedback. We have try to resolve all issues as you can find in attached file and revised manuscripts.

Round 2
Reviewer 1 Report
After a meticulous evaluation, I am pleased to note that the majority of the concerns raised in the initial review have been diligently addressed and rectified. As a result, I am confident that the manuscript is now prepared for publication, requiring no further revisions. My heartfelt appreciation goes to the dedicated researchers for their unwavering commitment and diligence in refining the paper to its current state.